# Effect of AuNPs and AgNPs on the Antioxidant System and Antioxidant Activity of Lavender (*Lavandula angustifolia* Mill.) from In Vitro Cultures

**DOI:** 10.3390/molecules25235511

**Published:** 2020-11-25

**Authors:** Paula Jadczak, Danuta Kulpa, Radosław Drozd, Włodzimierz Przewodowski, Agnieszka Przewodowska

**Affiliations:** 1Department of Plant Genetics, Breeding and Biotechnology, Faculty of Environmental Management and Agriculture, West Pomeranian University of Technology in Szczecin Słowackiego 17, 71-434 Szczecin, Poland; danuta.kulpa@zut.edu.pl; 2Department of Microbiology and Biotechnology, Faculty of Biotechnology and Animal Husbandry, West Pomeranian University of Technology in Szczecin, Piastów 45, 70-311 Szczecin, Poland; radoslaw.drozd@zut.edu.pl; 3Plant Breeding and Acclimation Institute in Bonin National Research Institute, Bonin 3, 73-009 Bonin, Poland; w.przewodowski@ihar.edu.pl (W.P.); a.przewodowska@ihar.edu.pl (A.P.)

**Keywords:** gold, silver, nanoparticles, antioxidant enzymes, free radical scavenging, shoot propagation, elicitation, micropropagation

## Abstract

The aim of this study was to determine the effect of gold and silver nanoparticles on the activity of antioxidant enzymes (ascorbate peroxidase (APX), superoxide dismutase (SOD), guaiacol peroxidase (POX), and catalase (CAT)), the free radical scavenging capacity, and the total polyphenol capacity of lavender (*Lavandula angustifolia* Mill.) cultivar “Munstead” propagated in vitro. In the experiment, fragments of lavender plants were cultivated in vitro on medium with the addition of 1, 2, 5, 10, 20, and 50 mg∙dm^−3^ of AgNPs or AuNPs (particle sizes 24.2 ± 2.4 and 27.5 ± 4.8 nm, respectively). It was found that the nanoparticles increase the activity of the antioxidant enzymes APX and SOD; however, the reaction depends on the NP concentration. The highest APX activity is found in plants propagated on media with 2 and 5 mg∙dm^−3^ of AgNPs. AuNPs significantly increase the APX activity when added to media with a concentration of 10 mg∙dm^−3^. The highest SOD activity is recorded at 2 and 5 mg∙dm^−3^ AgNP and AuNP concentrations. The addition of higher concentrations of nanoparticles to culture media results in a decrease in the APX and SOD activity. The addition of AuNPs to culture media at concentrations from 2 to 50 mg∙dm^−3^ increases the POX activity in comparison to its activity when AgNPs are added to the culture media. No significant influence of NPs on the increase in CAT activity was demonstrated. AgNPs and AuNPs increased the free radical scavenging capacity (ABTS^•+^). The addition of NPs at concentrations of 2 and 5 mg∙dm^−3^ increased the production of polyphenols; however, in lower concentrations it decreased their content in lavender tissues.

## 1. Introduction

True lavender (*Lavandula angustifolia*), classified in the mint family (*Lamiaceae*), is an evergreen shrub native to southern Europe, primarily the Mediterranean region. It is commercially cultivated in, among other places, France, Portugal, Poland, Spain, Hungary, the UK, Bulgaria, Australia, China, and the USA [1,2,3]. Essential oil with the characteristic aroma is one of the most important commercial components of true lavender, as it is used in the food, perfume, and the cosmetic industry [4,5,6]. Lavender oil is a mixture of chemical substances comprising primarily mono- and sesquiterpene compounds, exhibiting a complex therapeutic effect [7,8]. It is commonly used individually or as an additive in aromatherapy and folk and conventional medicine [9,10,11,12]. Numerous studies have confirmed the bactericidal and fungicidal properties of lavender oil toward *Staphylococcus aureus*, *Enterococcus*, *Aspergillus nidulans* and even *Trichophyton metangrophytes*, which is considered to be resistant to certain synthetic antibiotics [1]. In addition, lavender oil is characterized by analgesic, sedative, anticancer, and anti-inflammatory properties. It is a perfect agent to control digestive disorders and helps to combat skin disorders of various origins [10]. A study has also proven its effectiveness in alleviating depression symptoms [13].

Nowadays, the production of high-quality propagation materials, free of viruses and identical to parent plants, is ensured by biotechnological methods, including in vitro plant culture. Plants propagated in vitro are genetically uniform, which is of special importance for the production of secondary metabolites [14,15]. The large-scale production of lavender requires efficient in vitro propagation techniques to avoid the overexploitation of natural populations and allow the application of biotechnology-based approaches for plant improvement and the production of valuable secondary metabolites [16].

A series of studies have confirmed that medicinal plants propagated in vitro are characterized by different contents and compositions of essential oils [16,17,18,19]. The composition of secondary metabolites can also be changed using biotic and abiotic elicitors. These elicitors induce a series of physiological reactions in the plant and stimulate the production of secondary metabolites [20,21].

Nanotechnology has been applied in many fields of science and industry as a matter of specific qualities that have been identified by nanoparticles [22]. Nanometals are increasingly used in commercial products, including in the production of drugs and cosmetics and as transporting molecules, biosensors, or drug delivery agents [23,24,25]. They are also used in optoelectronics, nano-devices, and nanoelectronics. Their antibacterial properties have found application in medicine, water treatment, and food processing [26]. Resent research describe the wide use of nanosilver and gold in medicine [27,28,29,30].

They are valued for their unique physical and chemical properties (e.g., a higher surface to volume ratio, atypical surface structure, increased reactivity), and for their specific impact on biological tissues [31,32,33,34,35,36].

In recent times, particular attention has been paid to the use of nanotechnology in in vitro cultures. Nanoparticles have been mainly used as antimicrobial agents for sterilization and prevention of contamination of plant cultures in vitro [37]. In recent years, their influence on seed germination and plant growth has been increasingly studied [38,39,40]. The influences of copper (Cu), zinc (Zn), iron (Fe), titanium (Ti), and aluminium (Al) nanoparticles on plants are widely described among other things in *Phaseolus radiatus* [41], *Stevia rebaudiana* [42], *Sinapis alba*, *Lepidium sativum*, [43], *Cicer arietinum* [44], and *Nicotiana tabacum* [45]. However, the influence of nanoparticles of two metals, silver and gold, is most often studied. Ag and Au nanoparticles have been selected because of the features related to their commercial applicability—their low price, easy availability, and high stability in aqueous solutions—and the numerous studies indicating their low toxicity [46]. The effect of nanoparticles on plant development has been determined, such as the influence of silver nanoparticles on *Solanum tuberosum* [47] and *Eichhornia crassipes* [48] and gold nanoparticles on *Brassica juncea* [49], *Artemisia absinthium*, and *Vigna radiata* [50].

These nanoparticles are among the latest currently studied elicitors [51,52,53]. It has been proven that metal nanoparticles are highly reactive toward plants. Due to their small size, they can easily penetrate the cell membrane and accumulate inside the cell [54]. The properties of nanoparticles are strongly dependent on many factors, such as their method of preparation, the temperature at which they are prepared, and the surfactants used for capping [55].

Certain metal nanoparticles have a considerable impact on the physiological processes of plants, such as seed germination, growth, and metabolism [56,57,58,59], and scientific research indicates that their impact can be both positive and negative. The majority of studies describe that they exhibit a destructive influence that results in damage at the genomic level, chromosomal aberrations, cell growth inhibition, and programmed cell death-apoptosis [60,61,62]. Numerous studies have demonstrated the phytotoxicity of metal nanoparticles leading to the activation of the antioxidative system in plants [63,64]. Antioxidative enzymes, including guaiacol peroxidase (POX), superoxide dismutase (SOD), catalase (CAT), ascorbate peroxidase (APX), and alternative oxidase, and other low-molecular compounds in the form of polyphenols, vitamins, and minerals, are responsible for the inhibition of formation or the removal of excessive reactive oxygen species (ROS) from plant organisms [65,66].

The increase in the production levels of ROS under the influence of nanoparticles has been reported in many plants, such as *Oryza sativa* [67,68], *Alium cepa* [69], *Brassica juncea* [70,71], *Arabidopsis Tthaliana* [72], and *Nicotina tabacum* [73].

The oxidative stress induced by the metal nanoparticles may influence the production and qualitative and quantitative composition of secondary metabolites in plants [40,74]. The favorable influence of elicitation by metal nanoparticles on the production of biologically active compounds in plants has been confirmed by Zhang et al. [41,75] for *Artemisia anuua*, Sharafi et al. [42,76] for *Hypericum perforatum*, and Oloumi et al. [43,77] for *Glycyrrhiza glabra*. Silver and gold nanoparticles have been used as elicitors in cultures of *Prunella vulgaris* [78], *Stevia glycosides* [52], *Cucumis anguria* [79], and *Caralluma tuberculata* [80]. The latest scientific reports indicate significant changes in the metabolite profile of *Arabidopsis thaliana* under the influence of silver nanoparticles [81].

This paper is part of a cycle of research aimed at developing an efficient method of production of secondary metabolites, especially essential oils, by changing the profile of the secondary metabolites produced by *Lavandula angustifolia* plants from in vitro cultures. According to our previous studies, the addition to growth media of 1 to 50 mg∙dm^−3^ of AgNPs and AuNPs has a significant impact on plant development [82]. Our research has also shown that the addition of 10 or 50 mg∙dm^−3^ of AgNPs and AuNPs changes the composition of essential oils produced by lavender plants [83]. However, it is not clear to us whether this phenomenon is a result of stress or a change in the production of one of the endogenous plant growth regulators—e.g., ethylene, which modulates the endogenous production of other regulators. According to Akasaka-Kennedy [84], the addition of silver nanoparticles reduces the production of endogenous ethylene in *Brassica sp* tissues. Therefore, the main aim of the study is to answer the question of whether the addition of nanoparticles, which has been shown earlier, modifies their growth, affects the production of secondary metabolites, or activates ROS in plants grown on media in in vitro cultures. So far, there have been no scientific reports on the influence of AuNPs and AgNPs on the activation of the antioxidant system in *L. angustifolia* propagated in vitro.

The aim of the study was to analyze the influence of the addition to an in vitro culture of lavender of AgNPs and AuNPs on elements of the enzymatic antioxidant defences system, POX, APX, SOD, and CAT in plant tissue.

## 2. Results

### 2.1. Antioxidative Enzyme Activity

#### 2.1.1. APX

Silver nanoparticles added to the propagation medium at the lowest concentration used (1 mg·dm^−3^) did not affect the APX enzyme activity, which remained at the level of control plants (Figure 1). However, the addition of higher AgNP concentrations to the medium resulted in increasing the APX activity. The highest APX activity was determined in plants propagated on media with the addition of 2 and 5 mg·dm^−3^ of AgNPs, at 26.01 and 18.93 U·mg^−1^ DW, respectively. With the increased AgNP concentration in culture media in the range of 10 to 50 mg·dm^−3^, the enzyme activity decreased, but it was at a significantly higher level (*p* ≤ 0.05) compared to the activity of control plants. The addition of AuNPs to the culture media had a significant effect on the increase in APX activity in relation to the control, but only for AuNP concentrations equal to 10 mg·dm^−3^ (9.28 U·mg^−1^ DW).

#### 2.1.2. POX

The POX activity varied depending on the nanoparticle type applied. The POX activity was close to that of the control when plants were propagated on media with an AuNP addition at concentrations between 2 and 50 mg·dm^−3^ (Figure 1). A reduction in activity in relation to the control was observed solely for the lowest of the concentrations used (1 mg·dm^−3^). The propagation in media with the addition of AgNPs resulted in a dramatic reduction in the POX activity, relative to both the control as well as plants propagated on media with the addition of AuNPs. A particularly low POX activity was determined for plants propagated on media with an addition of 5 to 50 mg·dm^−3^ (0.30 to 0.57 U·mg^−1^ DW).

#### 2.1.3. SOD

The activity of SOD above the control was recorded for the concentrations 2 and 5 mg·dm^−3^ of AgNPs (866.22 and 1276.34 U·mg^−1^ DW, respectively). The addition of higher AgNP concentrations to the culture media (10, 20, 50 mg·dm^−3^) resulted in the decreased activity of the enzyme, thus it remained at the control level. Gold nanoparticles resulted in a decreased SOD activity (*p* ≤ 0.05) only at the highest concentrations applied—20 and 50 mg·dm^−3^ (407.36 and 312.28 U·mg^−1^ DW). The activity of SOD for the lowest AuNPs concentrations remained at the control level (Figure 1).

#### 2.1.4. CAT

The conducted study revealed that no significant impact of gold and silver nanoparticles could be found for the activity of catalase determined in the studied tissues (Figure 1).

### 2.2. Free Radical Scavenging Activity and Total Polyphenols Content

#### 2.2.1. The Total Polyphenol Content

Polyphenols content varied depending on the nanoparticle type and its concentration in the medium (Figure 2). Increasing polyphenol contents relative to the control were determined for lavender cultivated on media with the addition of 2 and 5 mg·dm^−3^ of AuNPs and 5 to 50 mg·dm^−3^ of AgNPs (0.214 and 0.217, respectively, and from 0.212 to 0.318 mg of TAE/100 gDW). On the other hand, the lavender cultivated on media with the addition of the highest AuNP concentrations (20 and 50 mg·dm^−3^) and lowest AgNPs concentrations (1 and 2 mg·dm^−3^) was characterized by significantly lower polyphenol contents.

#### 2.2.2. Free Radical Scavenging Activity (ABTS^•+^)

The presented outcomes suggest that gold and silver nanoparticles added to culture media in in vitro cultures have a positive impact on the free radical scavenging activity of true lavender (Figure 3). An exception here was the lowest AgNP concentration (1 mg·dm^−3^–197.77 μM AAE/100 gDW) and highest AuNP concentration used (50 mg·dm^−3^–158.86 μM AAE/100 gDW). The highest free radical scavenging capacity values were determined for plants cultivated on media with the addition of 5 mg·dm^−3^ of AgNPs and AuNPs, at 330.70 and 309.62 μM AAE/100 gDW, respectively. The addition of NPs in 1, 2, and 5 mg·dm^−3^ concentrations to the culture media resulted in a gradual increase in the antioxidant activity of lavender, whereas a reduced scavenging capacity was observed for higher concentrations (10, 20, and 50 mg·dm^−3^).

## 3. Discussion

Gold and silver nanoparticles are used on a large scale in many industries. Due to their antibacterial, anti-angiogenic, and anti-inflammatory properties, silver nanoparticles are used in antibacterial coatings for coating medical devices, wound and burn dressings, numerous cosmetic products, and food packaging [85,86,87]. Gold nanoparticles are used in molecular imaging, targeted drug delivery, gene therapy, cancer treatment, and radiation [88]. Prociak et al. [89] presented the potential use of gold and silver nanocolloids in cosmetic production. Microbiological tests performed in this study, showed that emulsion with silver and gold nanocolloids possessed satisfactory fungicidal properties and had a positive effect on the characteristics of the creams such as their consistency and absorption. Ag and Au nanocolloids can also be used in medicine, e.g., for the treatment of wounds, including the most difficult to heal burn and post-surgery wounds [90,91,92] or in medical imaging when diagnosing cancer [93,94]. It has been also confirmed the use of nanocolloids in the food industry, mainly in food packaging or prevention of contamination [95,96].

In view of the wide range of possibilities for the use of nanoparticles, it is important to determine the toxic properties of nanoparticles for humans and the environment. The results of most published scientific studies exclude the toxicity of nanoparticles [85,97,98]. However, there are also studies showing their adverse effects on living organisms and the environment, with this harmfulness depending on many factors, such as their physicochemical properties (shape, size, load) or coating agents [99,100]. Most of the studies show that nanoparticles are toxic to plants at higher concentrations. In addition, the type of target cells, tissues, or organisms or the type of test itself should be considered [100]. Therefore, it is vital to precisely determine the effect of nanoparticles on living organisms, including plants [100].

In response to oxidative damage, plants have developed a defense system which manifests itself, among other things, through the increased activity of antioxidant enzymes [101,102].

The activity of antioxidative enzymes may be a good marker of the toxic influence of the external environment on an organism. Mehrian et al. [103] studied the influence of AgNPs on the activity of antioxidant enzymes in *Lycopersicon esculentum*. It was found that, with the increase in the concentration of silver nanoparticles in the plant tissue medium, the activity of SOD, CAT, and POX in the shoots and roots increased. The increased activity of CAT, SOD, and POX was also observed in the callus tissue of sugar cane (*Saccharum spp.* cv CP-77,400) propagated on media with the addition of 20 to 60 AgNPs [104] (Barbasz et al.]. Iqbal et al. [105] carried out a study to investigate the effect of silver nanoparticles (AgNP) on the physiological, biochemical, and antioxidant parameters of wheat (*Triticum aestivum* L.) under heat stress conditions. It was found that the addition of AgNPs had a protective effect on plant tissues under stress conditions—wheat plants treated with AgNPs showed a significant increase in dry matter, with a simultaneous increase in SOD, POX, CAT, APX, and GPX activity under heat stress conditions. In the conducted study, we analysed the impact of the addition of gold or silver nanoparticles to culture media on the activity of CAT, APX, POX, and SOD antioxidative enzymes. The conducted study revealed a significant impact of the applied nanoparticles on the activity of all the tested antioxidative enzymes, with the exception of CAT. The observed effect was variable depending on the nanoparticle type and concentration. Lavender exposed to stress caused by AgNPs or AuNPs primarily activated the antioxidative system, consisting of APX and SOD. Increased activity of CAT, SOD, and POD due to Cu_2_O and Zn nanoparticles has also been confirmed for cucumber *Cucumis sativus* L. in the study of Kim and Lee [106]. These authors, using low concentrations of both kinds of nanoparticles, observed a significant increase in the activity of these three antioxidative enzymes. However, with the increased concentration of metals nanoparticles in the culture media (above 100 mg dm^−1^), the activity of these enzymes was markedly reduced. The concentration of the analyzed AgNPs which significantly reduced the activity of APX and SOD was 5 mg·dm^−3^. The experiment carried out by Gunjan et al. [107] examined the impact of gold nanoparticles on the activity of antioxidative enzymes in *Brassica juncea* seedlings. It was established that the glutathione reductase (GR), ascorbate peroxidase (APX), and glutathione peroxidase (GPX) activity increases with high AuNP concentrations (200 mg·dm^−3^), and that they are significantly higher than the CAT activity. These results suggest that the enzymatic complex in the form of APX, GPX, and GR forms part of the defensive mechanism against oxidative disturbance produced by nanometals in *Brassica juncea*, and that CAT does not have a substantial role in this case. In the study of Tripathi et al. [108], it was noted that silver nanoparticles (at concentrations from 100 to 300 µM) markedly stimulate superoxide dismutase and ascorbate peroxidase activity, while inhibiting the glutathione reductase and dehydroascorbate reductase activity in *Pisum sativum*.

Our study revealed that the true lavender antioxidative activity under in vitro cultures depends on the nanoparticle type and its concentration in the medium. *Lavandula angustifolia* is known for its health-promoting properties, resulting from the presence of antioxidative substances in its tissues (Miliasukas et al.) [109]. The chemical composition of plant metabolites and their antioxidative properties are influenced by various factors, including the i.a. cultivation method, growing location, climatic conditions, and plant genotype [2,110,111]. In our study, the free radical ABTS^•+^ scavenging analysis revealed a significant increase in the true lavender antioxidative activity after the addition of gold and silver nanoparticles to all culture media. This result is similar to the study of Chung et al. [79], where increased antioxidative activity was observed in transformed *Corylus avellana* roots under the impact of AgNPs. Fazal et al. [78] tested AuNPs and AgNPs as elicitors of *Prunella vulgaris* callus cultures and observed a marked increase in antioxidative activity.

Increased antioxidant activity may be associated, for example, with an increase in the content of substances with antioxidant activity, including polyphenols, in plant tissues. In our research we also examined the influence of the addition of AgNPs and AuNPs to the culture media of in vitro cultures on the polyphenol content. Polyphenols are secondary plant metabolites with strong antioxidants properties and are produced by plant cells for protective purposes against pathogens and also contribute in adaptation process to the external environment. Polyphenol compounds have also been suggested to have the ability to support activity of the cellular enzymatic antioxidative system, as well as modulate low-molecular antioxidant concentrations in the form of ascorbic acid (vitamin C) and alpha-tocopherol (vitamin E) in plant organisms [112]. In conducted study for determination of total phenolic compounds content in *Lavandula angustifolia* has been used the Folin-Ciocalteu assay. The sensitivity of this method can be affected by other cellular compounds such as sugars, proteins and lipids. However, depend on the plant organs, the total phenolic content can differ. Leaves of many plants are especially rich in this compounds and eventual interference other co-extracted substances do not affect the quality of assay [113]. Despite its limitations, Folin-Ciocalteu method is widely used to determine the content of polyphenols in the tissues of plants grown in in vitro cultures [114]. It was used, inter alia, by Tian et al., (2018) [115] determining the content of polyphenols in *Atropa bellladonna* tissues treated with Mn_2_O_3_ NPs, callus tissues of *Caralluma tuberculata* treated with AgNPs [80] or *Stevia rebaudiana* shoots treated with ZnONPs [42]. Previous research has stated that the influence of metal nanoparticles on the content of polyphenols in vegetable plants is variable. An increased content of polyphenols under the impact of AgNPs has been determined for *Bacopa moonnieri* [116] and *Solanum tuberosum* [117] and a reduction for castor [118]. We obtained similar results in our research. The polyphenol content was higher than in control plants when the plants grew on media with the addition of 10–50 mg·dm^−3^ AgNPs and 2–5 AuNPs. The study of Jamshidi and Ghanati [119] provided evidence of the significant effect of silver nanoparticle elicitation on the increase in phenol and flavonoid contents in hazel plants. In their experiment, they utilized low AgNP concentrations (2, 5, 10 mg·dm^−3^), indicating their clear impact on the content of the tested antioxidants in cell cultures. Ghorbanpour and Hadian [120] showed the significant influence of MWCTs (multi-walled carbon nanotubes) in the culture of the callus tissue of *Satureja khuzestanica* on the total flavonoid content in callus extracts for all MWCNT concentrations used. However, the analysis of the polyphenol content showed that its increase was observed only for some of the concentrations applied (50, 100, and 200 μg mL^−1^ MWCNTs). Their experiment further revealed that the maximum values for flavonoids and phenols were obtained using a MWCNT concentration of 100 μg·mL^−1^. In the present study, the nanoparticle concentrations used in culture media were lower. Tian et al. [115] demonstrated the significant impact of Mn_2_O_3_NPs at the concentration 25 mg·dm^−3^ on the increase in the alkaloid content in *Atropa balladona* L. tissues.

Production of bioactive secondary metabolites is generally associated with plant defense mechanisms. However, the mechanism of action of nanoparticles and their influence on ROS activation and the production of secondary metabolites is still poorly understood. According to Saha and Gupta (2018) [121], who studied the influence of silver nanoparticles on the development of *Swertia chirata* in in vitro cultures, ROS generated in presence of AgNP triggered an array of antioxidative enzymes which later on balanced the ROS content in the treated plant system. These enzymes also stimulated the shoot proliferations and helped in maximization of generation of number of shoots per explant considered. According to the authors, the reduction of ethylene production due to AgNPs may have a stimulating effect on plant development. Amir et al., (2019) [80] suggest that NPs might act as signal compounds, those like in nature to the chemical elicitors and influence cellular growth and secondary metabolism. Upon exposure to application of NPs, plant cell goes through a series of events in a cascade manner, resulting in oxidative outburst and generation of reactive oxygen species (ROS) in the surrounding environment of the plant cell. The ROS in turn can damage cell membrane and nuclei. To cope with the intense stress situation and to scavenge the ROS, plants activate their metabolic pathways including the notable mitogen-activated protein kinase (MAPK) pathway. MAPK activation propels the plant antioxidant elements to come in contact with ROS in a cascade fashion. Several studies have shown that nanoparticles, especially AgNPs, affect the up-regulation of genes related to the response to abiotic stress, including the production of enzymes such as SOD, CAT, AXP [122,123]. Nair and Chung, (2014) [124] when examining the toxicity of AgNPs on rice, found a significant up-regulation of SOD gene. In their opinion, induction of SOD gene in response to AgNPs stress might be to maintain the redox homeostasis of the cell.

Our results suggest that gold and silver nanoparticles added to in vitro culture media have a significant impact on the antioxidative potential (free radical scavenging capacity) and activation of antioxidative enzymes of in vitro true lavender cultures. The defence mechanism of lavender exposed to stress caused by AgNPs and AuNPs primarily consists of the APX-SOD enzymatic complex, with CAT remaining insignificant in this case. It can additionally be stated that AgNPs at the concentration of 5 mg·dm^−3^ result in the highest antioxidative activity of APX and SOD, and higher concentrations in the culture medium inhibit their activity. AgNPs have a more toxic impact on lavender plants cultivated in vitro than AuNPs. Silver and gold nanoparticles added to culture media have significant impacts on the total polyphenol contents in true lavender cultivated in vitro and its free radical scavenging capacity ABTS^•+^, however this depends on the applied nanoparticle concentration.

## 4. Materials and Study Methods

### 4.1. Plant Culture under In Vitro Conditions

The study material comprised *L. angustifolia* cultivar “Munstead” plants growing under in vitro conditions. The explants used for the experiment were single-node shoot fragments and apical shoot fragments with several leaves and the apical meristem. They were placed on the Murashige and Skoog (MS) media [125] with the addition of 2 mg dm^−3^ of kinetin and 0.2 mg dm^−3^ of indole-3-acetic acid [126] and 1, 2, 5, 10, 20, or 50 mg dm^−3^ of AgNPs or AuNPs. The medium pH was set at 5.7 using 0.1 M solutions of HCl and NaOH. The culture medium was hardened with 7 g·dm^−3^ of agar and was subject to a 20-min sterilization in an autoclave at 121 °C and a 1 ATM pressure. Propagation was carried out in jars of 200 mL volume, containing 20 mL of medium. Eight explants were placed in each culture vessel. The experiment was established in 10 replications for each type and concentration of the nanoparticles used. The jars were placed for 28 days in a growth chamber at a temperature of 24 °C and relative air humidity of 70–80%. The cultures were illuminated with fluorescent light at an intensity of 40 PAR (μE·m^−2^·s^−1^) for 16 h per day. After 28 days of culture, the study material was collected and subjected to analyses.

### 4.2. Nanoparticles

In the experiment, aqueous solutions of AuNPs measuring 24.2 ± 2.4 nm and AgNPs measuring 27.5 ± 4.8 nm were used. The aqueous suspensions were synthesized using the methods of Turkevich et al. [127] and Liu et al. [128] with modified synthesis conditions and two-stage microwave-convection heating. For this purpose, aqueous mixtures of 3.5 mM sodium citrate with 7.0 mM tetrachloroauric acid (HAuCl_4_) and 7.0 mM silver nitrate (AgNO_3_), respectively, were prepared. Once their spectra were plotted with a UV-Vis Epoch Microplate Spectrophotometer (BioTek, Winooski, VT, USA), the optical density of the fractions obtained was adjusted to a common DEV value using the spectra absorbance maxima (λ_max_ = 520 nm for gold colloid and λ_max_ = 445 nm for silver colloid). The similarities in the morphology, shape, and size of the synthesized and prepared AuNPs and AgNPs were assessed through analyzing their images obtained using a transmission electron microscope (TEM) JEM-2100 (JOEL ltd., Tokyo, Japan) (Figure 4).

### 4.3. Antioxidant Enzymes Activity Assay

#### 4.3.1. Sample Preparation

All the steps of the protein extraction were carried out at 4 °C on a cooled plate. Fresh leaf samples weighing approximately 0.2 g were previously powdered in liquid nitrogen using a cold mortar and pestle. The obtained powder was resuspended in 1 mL of cold 50 mmol L^−1^ phosphate buffer (pH 7.0) enriched with 1 mmol L^−1^ of phenylmethylsulfonyl fluoride, 2 mmol L^−1^ of EDTA, and 1% polyvinylpyrrolidone. The resulting solution was centrifuged at 15,000 rpm for 10 min at 4 °C. The extracts were immediately used for the determination of the enzyme activity or stored at −80 °C for further analysis.

#### 4.3.2. Guaiacol Peroxidase Assay (POX)

The activity of POX was determined according to the protocol of Chance and Maehly [129], which was adapted to a microplate reader, by monitoring the rate of tetraguaiacol formation from guaiacol by an increase in absorbance at 470 nm (ε = 26.6 mM^−1^ cm^−1^). One unit of POX is defined as the amount of enzyme that forms 1 µmol of tetraguaiacol per minute. The results were expressed in units per mg of the extracted protein.

#### 4.3.3. Ascorbate Peroxidase Assay (APX)

The activity of APX was measured according to the protocol of Nakano and Asada [130] by monitoring the decrease in absorbance at 290 nm (ε = 2.8 mM^−1^ cm^−1^). The measurements were performed using a microplate reader. One unit of APX is defined as the amount of enzyme that oxidizes 1 µmol of ascorbate per minute. The results were expressed in the units per mg of the extracted protein.

#### 4.3.4. Dismutase Assay (SOD)

The total SOD activity was determined based on the inhibition of the photochemical reduction of nitroblue tetrazolium (NBT) according to the protocol of Beauchamp and Fridovich [131], which was adapted to a microplate reader. One unit of SOD is defined as the amount of enzyme that inhibits the NBT reduction by 50%. The results were expressed in units per mg of the extracted protein.

#### 4.3.5. Catalase Assay (CAT)

The activity of CAT was determined according to the protocol of Li and Schellhorn [132] based on the rate of decomposition of hydrogen peroxide by measuring the decrease in absorbance at 240 nm (ε = 43.6 mM^−1^ cm^−1^). One unit of CAT is defined as the amount of enzyme that decomposes 1 µmol of hydrogen peroxide per minute. The results were expressed in units per mg of the extracted protein.

### 4.4. Total Phenol Content and Free-Radical ABTS^•+^ Scavenging Ability Assay

#### 4.4.1. Tissue Extract Preparation

A total of 100 mg of tissue (previously frozen at −80 °C) was powdered in liquid nitrogen using a cooled laboratory mortar and pestle. The obtained powder was transferred to plastic tubes, mixed with 5 mL of cooled methanol, and left for 1 h in the dark. During extraction, the samples were mixed periodically. Then, the samples were centrifuged at 15,000 rpm and the obtained extract was used for further analysis.

#### 4.4.2. Total Polyphenol Content Assay

The total content of polyphenolic acid in the tissue extracts was determined using Folin-Ciocalteu reagent according to the method of Anastasiadi et al. [133] which was modified to a microplate reader scale. The concentration of polyphenolic compounds was expressed as the mg of tannic acid equivalent (TEA) per 100 g of dry weight of sample (mg TAE/100 gDW).

#### 4.4.3. Free Radical ABTS^•+^ Scavenging Ability Assay

The free radical scavenging activity was assessed in the tissue extracts using the cation radical discoloration of the 2,2′-azino-bis (3-ethylbenzothiazoline-6-sulfonic acid) diammonium salt ABTS^•+^ assay adapted to a microplate reader scale, according to the method of Shi et al. [134]. Ascorbic acid (AAE) was used as a standard for calibration. The results were expressed in the μM of ascorbic acid equivalent per 100 g of dry weight of sample (μM AAE/100 gDW).

### 4.5. Statistical Analysis

The experiment was performed in three replications. An analysis of variance was performed, followed by Tukey’s test (*p* ≤ 0.05). Homogenous groups were labeled with successive letters of the alphabet.

## Figures and Tables

**Figure 1 molecules-25-05511-f001:**
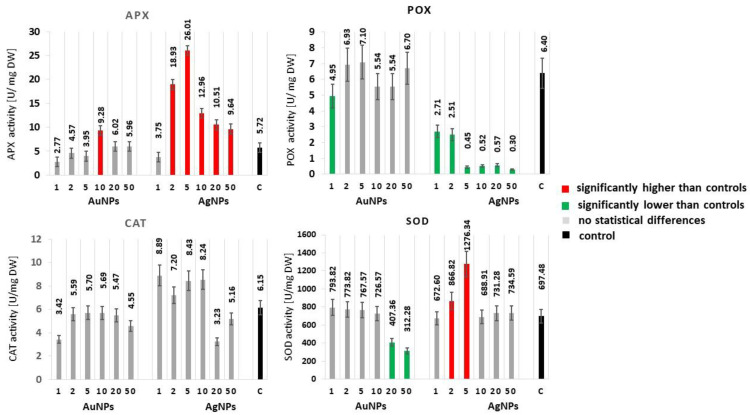
Effect of AgNPs and AuNPs (1–50 mg∙dm^−3^) on the antioxidant enzymes activity in *Lavandula angustifolia*. a-ascorbate peroxidase (APX), b-guaiacol peroxidase (POX), c-catalase (CAT), and d-superoxide dismutase (SOD). Values represent the means of three replications ± SE. Different colors of bars indicate significant statistical differences between the treatments and control samples according to Tukey’s Test (*p* < 0.05).

**Figure 2 molecules-25-05511-f002:**
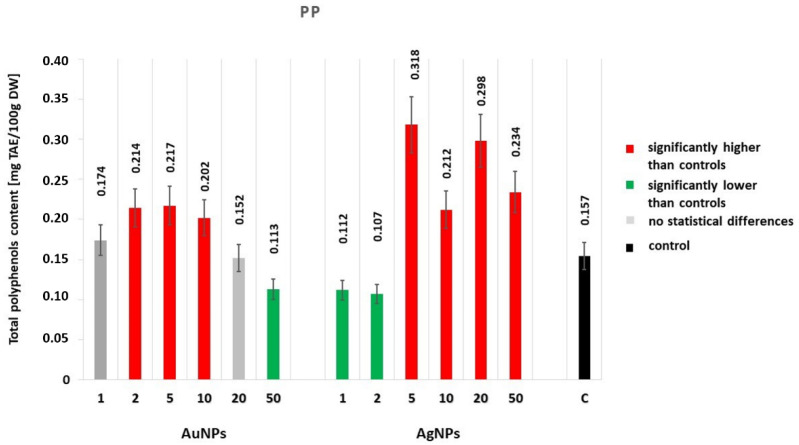
Effect of AgNPs and AuNPs (1–50 mg·dm^−3^) on the total polyphenol content (PP) of *Lavandula angustifolia* (±SE). Different colors of bars indicate significant statistical difference between the treatments and control samples according to Tukey’s Test (*p* < 0.05).

**Figure 3 molecules-25-05511-f003:**
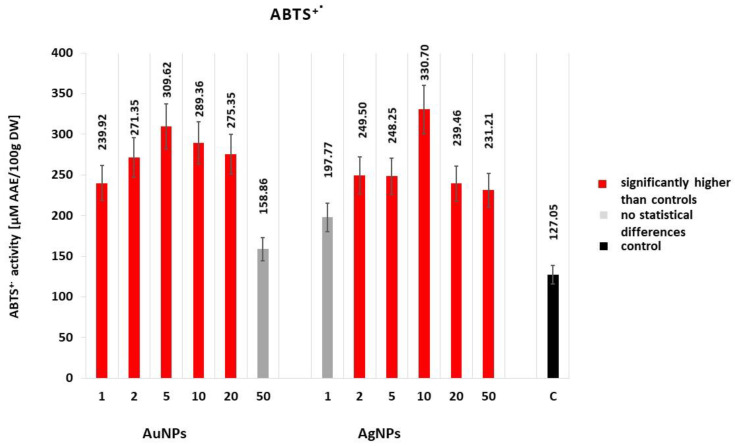
Effect of AgNPs and AuNPs (mg∙dm^−3^) on the antioxidant activity of *Lavandula angustifolia* in vitro cultures measured by the ABTS^•+^ radicals scavenging assay. Values represent the means of three replications ±SE. Different colors of bars indicate significant statistical differences between the treatments and control samples according to Tukey’s test (*p* < 0.05).

**Figure 4 molecules-25-05511-f004:**
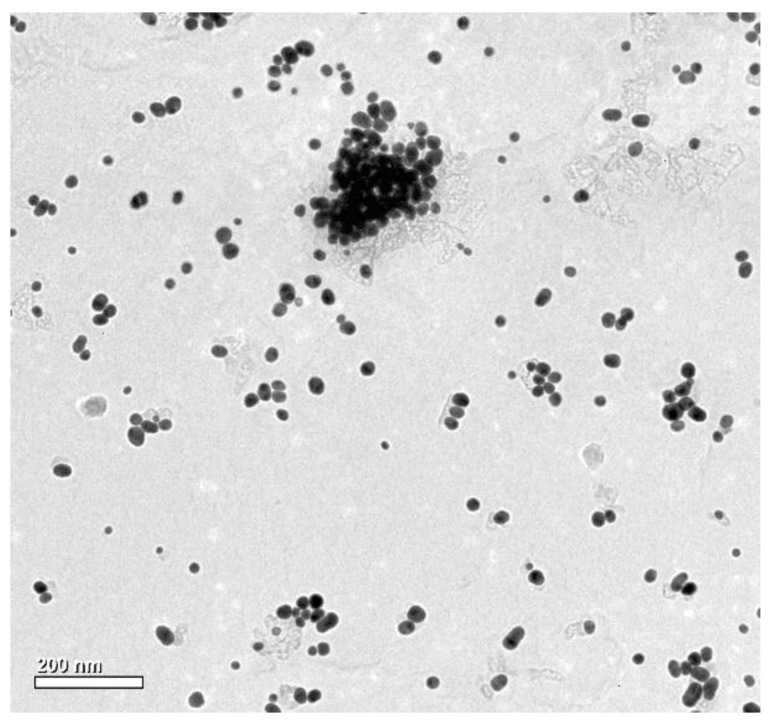
TEM image of AgNPs suspended in water; scale bar: 200 nm.

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
