# Peer review of "Effect of AuNPs and AgNPs on the Antioxidant System and Antioxidant Activity of Lavender (Lavandula angustifolia Mill.) from In Vitro Cultures"

_molecules, 2020, doi:10.3390/molecules25235511_

Round 1

Reviewer 1 Report

P. Jadczak and coauthors presents data on the effect of AuNPs and AgNPs on the enzymatic activity of the main enzymes of the antioxidant system, polyphenol levels, and free radicals scavenger system capacitance. The work was carried out by adequate methods and the results obtained are valuable. However, the work can be published after major revisions only.

The main points:

There are no data on the dose-dependent effect of nanoparticles on the physiological state of the plants.

The “Introduction” should more directly lead the readers to the aim of the study. In manuscript, the aim is not fully formulated, it does not contain an explanation for what the parameters of the antioxidant system are measured. What the authors want to achieve with their results, which are undoubtedly reliable?

The “Discussion” is a retelling of the results. The authors do not discuss whether AuNPs and AgNPs, at low concentrations, can be considered as the elicitors.

The introduction and discussion should be revised.

The minor points

In the methods, 4.2. Nanoparticles: “Once their spectra were plotted with a UV-Vis…(Fig. 1)”; in the Results: “…APX enzyme activity, which remained at the level of control plants (Figure 1).” A Figure demonstrated nanoparticles properties is absent.

Line 175: “glutathione reductase dehydroascorbate reductase” should be changed to “glutathione dehydroascorbate reductase”.

Author Response

Authors’ Response to the Reviewers’ Comments

Journal: Molecules

Manuscript:  molecules-929720

Title of Manuscript: Effect of AuNPs and AgNPs on Antioxidant System and Antioxidant Activity of Lavender (Lavandula angustifolia Mill.) from In Vitro Culture

Authors: Paula Jadczak, Danuta Kulpa, Radosław Drozd, Włodzimierz Przewodowski  and Agnieszka Przewodowska

We appreciate the time and efforts by the editor and referees in reviewing this manuscript. We have addressed all issues indicated in the review report, and believed that the revised version can meet the journal publication requirements.

Reviewer 2 Report

I like this kind of rather unusual paper and I encourage publication but after mandatory revision and re-review.

The paper suffers from a very short nature where introduction and discussion is limited. Aspects such as why these nanoparticles (see review in IoP nanotechnology and other recent papers by Matharu including recent ones in nanomaterials, Medical Dev and Sensors) should be fully discussed. 

Health and Safety should be fully discussed - why Ag-based?

How are these delivered? via patches? Relevant methods need be discussed - see recent work by Alenezi et al. 2019 in Appl Phys Rev and by Ahmed et al. in Biotech Adv 2020.

Accuracy is a major factor - check accuracy in Figs 1-3 and has error estimation been done (not just relative to a control), accuracy to so many decimals valid?

There are no micrographs of the nanoparticles and how they sit in the material - can the material be processed in reality and what are its properties, can it be mixed with a carrier? Delivery is so important if aimed at the healthcare market.

Increase antimicrobial testing, comparing with standards at all points.

Author Response

(The authors gave the same response as above.)

Round 2

Reviewer 1 Report

The authors have significantly revised the text of the article. They made technical corrections and provided detailed answers to questions. In the present form, the article can be recommended for publication.

Author Response

Thank you very much for your comment.

Reviewer 2 Report

The authors have not addressed the reviewers comments well. Most are vague responses, not leading to revisions.

I have given the authors a way forward too, in a high quality journal like Molecules we must have wider discussion, comparison with literature etc.etc. This has to be done.

Please revise, ADOPTING WHAT I HAVE SUGGESTED.

Author Response

Thank you very much for the comment.

I am attaching the fie with responses,

Kind regards,

Paula 
